# RECURRENT DRAFTER FOR FAST SPECULATIVE DECODING IN LARGE LANGUAGE MODELS

## ABSTRACT

We present Recurrent Drafter (ReDrafter), an advanced speculative decoding approach that achieves state-of-the-art speedup for large language models (LLMs) inference. The performance gains are driven by three key aspects: (1) leveraging a recurrent neural network (RNN) as the draft model conditioning on LLM's hidden states, (2) applying a dynamic tree attention algorithm over beam search results to eliminate duplicated prefixes in candidate sequences, and (3) training through knowledge distillation from the LLM. ReDrafter accelerates Vicuna inference in MT-Bench by up to 3.5x with a PyTorch implementation on Nvidia H100 GPUs. To demonstrate its practicality in production environments, we integrate ReDrafter into TensorRT-LLM, reaching up to 2.5x speedup on H100 GPUs. We also validated its effectiveness for on-device applications by implementing the approach in MLX and benchmarking performance on Metal GPUs in Apple Silicon chips, achieving up to 2.3x speedup. We summarize our experimental results in Figure 1.

## 1 INTRODUCTION

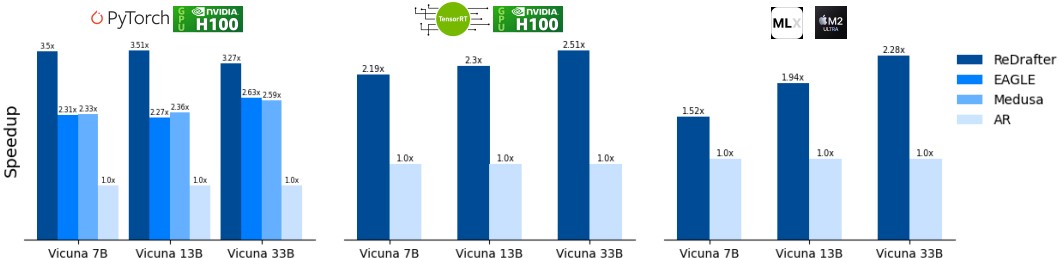

Figure 1: Inference speedup on MT-bench for Vicuna models are shown for three ReDrafter implementations: (left) PyTorch on Nvidia H100 GPU, (mid) TensorRT-LLM on Nvidia H100 GPU, (right) and MLX on Apple's M2 Ultra Metal GPU. We compare with EAGLE (Li et al., 2024a), Medusa (Cai et al., 2024), and auto-regressive in PyTorch implementations.

Speculative decoding (Leviathan et al., 2023; Spector & Re, 2023; Cai et al., 2024; Bhendawade et al., 2024) has been investigated as a promising technique to accelerate large language model (LLM) inference (Brown et al., 2020; Touvron et al., 2023; Achiam et al., 2023; Anil et al., 2023a;b; Gunter et al., 2024). It uses smaller, more efficient models (often referred to as draft models) to predict candidate sequences, which are then verified by the LLM. The underlying idea is to allow the LLM to focus on validating those candidates rather than generating every token sequentially. This approach helps to mitigate the bottleneck of memory bandwidth by reducing the need for repeated passes through the LLM. Since the draft model can introduce overhead, the reduction in LLM calls must be sufficient to offset this cost in order to achieve a net speedup.

Recently, Medusa (Cai et al., 2024) achieved significant speedup using small draft heads attached to LLM's hidden state, instead of maintaining a separate draft model. However, Medusa necessitates multiple draft heads with distinct parameters to indicate predictive positions. Its independent prediction mechanism does not leverage the sequential structure, resulting in limited predictive accuracy and an exponentially large set of feasible candidate token sequences.

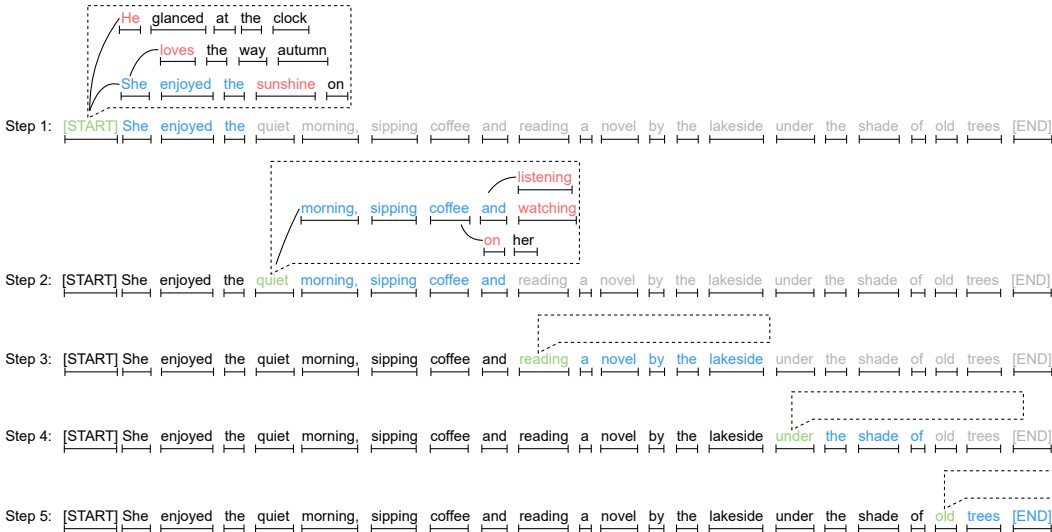

Figure 2: ReDrafter decoding process illustration. At each inference step, LLM generates an initial token (highlighted in green). Following this, the draft model performs a beam search (indicated by the dashed box) using the guaranteed token and the last-layer hidden states from the LLM as inputs. The LLM then verifies the beam, and the longest accepted prefix (highlighted in blue) is appended to the context. This allows each step to accept multiple tokens through a single forward pass from the LLM. The process repeats until the sequence concludes. ReDrafter ensures that the tokens it generates are identical to those produced by the LLM (marked as gray).

In this paper, we introduce the Recurrent Drafter (ReDrafter), for fast LLM inference. Figure 2 illustrates its generative process. ReDrafter's performance gains are driven by three key aspects: (1) Using a recurrent neural network (RNN) (Mikolov et al., 2010) conditioned on the LLM's hidden states as the draft model harnesses local temporal dependencies, improving the accuracy of the drafter's predictions and effectively converting computational resources into speedups. (2) By utilizing beam search to explore multiple candidate sequences and applying a dynamic tree attention algorithm to eliminate duplicated prefixes among the candidates, we significantly reduce computational overhead. (3) Training through knowledge distillation (Zhou et al., 2023) from LLMs improves the alignment of the draft model's predictions with those of the LLM, effectively transferring the computational load from inference time to training time.

In a PyTorch implementation, ReDrafter accelerates Vicuna inference by up to 3.5x compared to the autoregressive method in MT-Bench on Nvidia H100 GPUs, achieving state-of-the-art performance. Additionally, we demonstrate the effectiveness of ReDrafter in production environments through two key use cases. The first is a production-level deployment on TensorRT-LLM, designed to manage high traffic with long context lengths. In this implementation, tensor parallelism and continuous batching are utilized to ensure the system efficiently handles larger volume of requests while maintaining low latency. In this scenario, ReDrafter achieves up to 2.5x speedup on the MT-bench benchmark. The second use case focuses on an on-device approach using MLX on Metal GPUs within Apple Silicon chips. Despite the limited compute resources in this setup, we observed a memory bottleneck. ReDrafter effectively mitigates this bottleneck, resulting in up to 2.3x speedup, demonstrating its capability to optimize performance in resource-constrained environments.

## 2 RELATED WORK ON SPECULATIVE DECODING

It is widely recognized that LLM generation is constrained by the memory bottleneck. Speculative decoding mitigates this bottleneck by increasing computational intensity, utilizing a smaller draft model to locally predict probable future tokens.

Recently, draft model design has undergone numerous iterations. Leviathan et al. (2023); Chen et al. (2023); Spector & Re (2023); Sun et al. (2024) choose to use separate draft models detached from

LLMs. This is a handy choice when there is an off-the-shelf model closely approximates the LLM. For example, a common approach is to use a smaller variant from the same model family as the draft model. If no off-the-shelf candidate is available, the draft model must be trained separately from the LLM, with efforts focused on aligning it as closely as possible to the LLM. Additionally, deploying two separate models adds complexity to their integration within a unified serving system.

Another thread of approaches employs a unified strategy by attaching the draft model to the LLM, making them dependent (Stern et al., 2018; Santilli et al., 2023; Cai et al., 2024). This is an efficient design choice when the draft model is not intended for standalone use, allowing it to leverage additional computational resources for local prediction by conditioning on the LLM. Among these methods, Cai et al. (2024) proposes to use $T$ independent draft heads to predict next $T$ tokens, utilizing GPU's extra parallel computing power. This excessive computational effort may not yield proportional speedups, as independent predictions become less accurate as $T$ increases, resulting in suboptimal predictive accuracy and a lower acceptance rate from the LLM. An alternative approach is to enhance prediction accuracy by incorporating recurrence into the draft model to capture local dependencies (Bhendawade et al., 2024; Li et al., 2024a;b). However, the reduced parallelism due to recurrence leads to lower GPU utilization, introducing overhead that diminishes speedup gains, even when the acceptance rate is substantially higher.

ReDrafter uses a lightweight RNN as the draft model to predict upcoming tokens in the sequence. It allocates computational resources to beam search, resulting in a higher acceptance rate. The intensity of the beam search is controlled by the beam width and length, which can be adjusted based on hardware capabilities and specific implementations. ReDrafter applies knowledge distillation (Xia et al., 2023; Miao et al., 2023; Liu et al., 2023; Zhou et al., 2023) from LLMs, improving inference time efficiency by investing more resource in training time. Our empirical results reveal that ReDrafter utilizes compute more effectively compared to previous methods, delivering state-of-the-art speedups across various implementations and hardware platforms.

## 3 ReDrafter

### 3.1 Draft Model

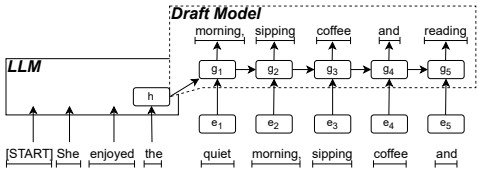

Figure 3: Draft model takes LLM's last hidden state $h$ as input to forecast next few tokens. For brevity, we omit model parameters and LLM hidden states before $h$.

We outline the formulation of the draft model in Figure 3. Similar to the Medusa approach, we use the last layer's output of the transformer from the LLM as input to the draft model. Additionally, we incorporate the embeddings of historical tokens as recurrent inputs to the draft head.

We use the standard RNN design to predict the next token. For instance, considering the context *"She enjoyed the"*, once LLM generates the token *"quiet"* with last layer's output $h$, the draft model uses the embedding of token *"quiet"* to update its RNN hidden state and combine with output $h$ to predict the next token *"morning"*. This strategy is recurrently applied for subsequent tokens. In this paper, we opt for a simple recurrent design to model the connections among the shared draft heads, deferring more complex model choices to future investigations. In particular, we initialize the hidden state of the draft model as $g_1 = [s_1, h]$, where $s_1 := e_1$ is the embedding of the last token that LLM produced. To predict the $t$th token using the draft model, we first update its hidden state,

$$g_t = [s_t, h], \quad s_t = f(U s_{t-1} + W e_t + b),$$

where $f$ is the activation function and $U$, $W$ and $b$ is the parameter for the RNN (Mikolov & Zweig, 2012). We only use one layer RNN to make the model simple. Then we apply a few layers of MLPs with skip connections, followed by a standard softmax layer at the end. Since the parameters of the draft heads are shared, the number of parameters remains constant even if the model is trained to predict more than one token.

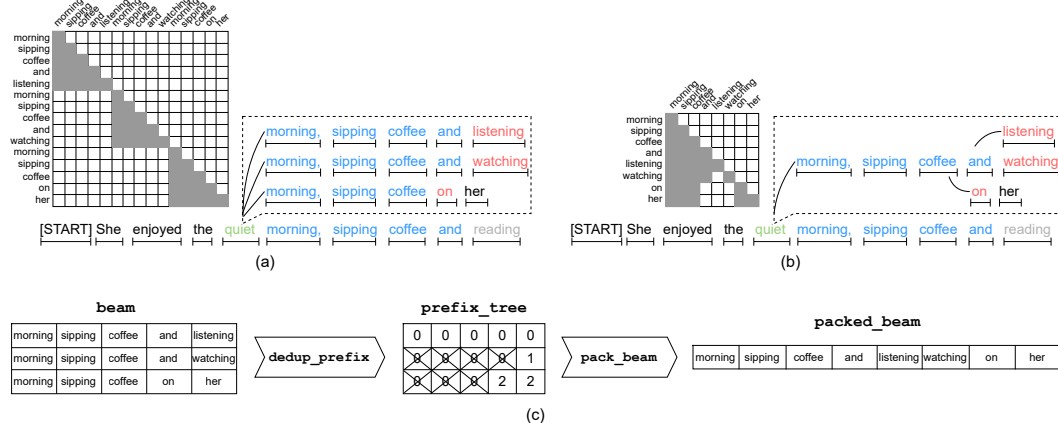

Figure 4: Beam search produces three candidate sequences. (a) Without dynamic tree attention, after flattening the tokens across all candidates, 15 tokens need to be sent to the LLM for verification. (b) With dynamic tree attention, we can trim the shared prefix, reducing the total to 8 tokens after flattening. We adjust tree attention masks accordingly to reflect token dependencies. (c) An illustration for building the dynamic tree attention for batch size equals to 1. (Extending to batch size larger than 1 is straighforward.) We use tensor-based, GPU-friendly algorithm to pack the beam into a "packed beam". The attention masks can be processed accordingly in a GPU-friendly way.

## 3.2 BEAM SEARCH

The draft model triggers the beam search algorithm during inference. This allows the model to explore a variety of possible continuations given the context, ranking them by probability while keeping track of multiple potential sequences. It helps maintaining a balance between diversity and optimality in candidate generation.

The parameter beam width refers to the number of draft token sequences. A larger beam width increases the likelihood that the sequence with the longest acceptable prefix is included in the beam. This allows the LLM to accept more tokens in each decoding step, reducing the overall number of decoding steps required—or, equivalently, decreasing the number of calls to the LLM. While a wider beam requires more FLOPs for beam search and for LLM verification, a powerful GPU can process these FLOPs in parallel, minimizing the increase in wall time.

## 3.3 DYNAMIC TREE ATTENTION

Beam search returns draft candidates with shared prefixes. For examples, in Figure 4(a), the second candidate *"morning sipping coffee and watching"* and the third candidate *"morning sipping coffee on her"* share a prefix *"morning sipping coffee"*. Sending those duplicated tokens to LLM result in computation overhead. As a result, we remove those shared prefixes, revealing a tree structure over beam search results. When sending de-duped candidate to LLM for verification, we modify the attention mask to reveal dependencies among tokens (Figure 4(b)). We refer this mechanism as "dynamic tree attention".

The use of a tree structure to save computation resembles approaches seen in Miao et al. (2023); Spector & Re (2023); Cai et al. (2024), where similar methods were employed to avoid redundant calculations for shared prefixes. However, unlike the use of tree structures mentioned above, we must determine ours *dynamically* as it relies on individual beam search results at runtime. A standard trie-based algorithm is slow as it is difficult to be parallelized. We notice that a unique property of our problem is that all candidate sequences have the same length. With this insight, we discovered that our dynamic tree attention can be efficiently constructed using standard tensor operators, a critical aspect for leveraging modern-day accelerators with little overhead. This leads to further acceleration of LLM inference in speculative decoding.

The key observation is that we can find all shared prefixes in the beam search result using standard tensor operations without building a prefix trie. We have developed functions called `dedup_prefix` to identify shared prefixes, and another function `pack_beam` to compress a beam into a deduped, *"packed beam"* (illustrated in Figure 4(c)). We provide the pseudo-code for `dedup_prefix` in the Appendix A.1 to provide a glimpse of how this tensor-based algorithm works. Subsequent operations can be similarly designed to use the prefixes found here to construct dynamic tree attention. This leads to further acceleration of LLM inference in speculative decoding. The use of dynamic tree attention is not limited to ReDrafter. It can also be used in detached speculative decoding approaches while a separate draft model performs beam search and then apply dynamic tree attention.

## 3.4 Speculative decoding with ReDrafter

We briefly describe the steps of using ReDrafter for speculative decoding. In each generation step during inference, ReDrafter alternates between using the draft model to generate tokens and calling the LLM to verify and accept them. We start with all previously generated tokens with the last hidden state. We employ beam search to generate a beam, consisting of a set of candidate sequences. Subsequently, dynamic tree attention is applied to flatten and compress the beam into a packed beam, while formulating an appropriate attention mask. LLM then proceeds with a forward pass to compute the log probabilities for all proposed tokens. Then, we select the best candidate with the longest accepted prefix. The selection method can range from a greedy approach (aka. token matches), to rejection sampling. Simultaneously, LLM provides hidden states and the initial token for the next draft model call. We append accepted tokens to the end of previously generated tokens and run the next iteration until the stopping criteria are met. ReDrafter guarantees the generated sequence matches LLM's output.

## 3.5 ReDrafter training with knowledge distillation

ReDrafter's efficiency is optimized when all candidate tokens are accepted. That is, draft model make the same prediction as the LLM within its prediction range $T$. A natural loss function is the KL divergence:

$$\min_{p_{\text{draft}}} \ \text{KL}(p_{\text{llm}}(y_{1:T})|p_{\text{draft}}(y_{1:T})) = \min_{p_{\text{draft}}} \ \mathbb{E}_{p_{\text{llm}}(y_{1:T})} - \log p_{\text{draft}}(y_{1:T}) \tag{1}$$

This implies that, instead of using the ground truth token as the label for the draft model, we should utilize the probability predictions from the LLM, in a manner similar to traditional knowledge distillation approaches (Kim & Rush, 2016). At each position $t$ of a training sequence, we sample $\widehat{y}_{t+1:t+T}$ conditioning on $y_{1:t}$ from LLM and optimize the following empirical loss

$$\min_{p_{\text{draft}}} L_{\text{distill}} = \min_{p_{\text{draft}}} \ \sum_t - \log p_{\text{draft}}(\widehat{y}_{t+1:t+T}|y_{1:t}). \tag{2}$$

Other speculative decoding methods, like Medusa2 (Cai et al., 2024), also incorporate knowledge distillation. In contrast, ReDrafter only backpropagates through the draft model, keeping the LLM unchanged to ensure the decoding results remain consistent. Additionally, we apply distillation locally by having the LLM predict the next $T$ tokens using the ground truth tokens as context.

## 4 Experiment

We conduct experiments in experimental and production-ready environments, using Vicuna 7B, 13B, 33B models as base LLMs. First, using PyTorch, we compare ReDrafter with state-of-the-art speculative decoding methods on an Nvidia H100 GPU in Section 4.1. Next, we validate ReDrafter's performance gain in a production-ready environment on an inference server using TensorRT-LLM on H100, leveraging tensor parallelism and continuous batching in Section 4.2. Moreover, we investigate the on-device deployment of ReDrafter on Metal GPU using MLX, demonstrating its ability to accelerate on-device inference with limited computational resources in Section 4.3.

Additionally, we conduct ablation studies using PyTorch. In Section 4.4.1, we explore ReDrafter's performance trade-offs when adjusting beam width and batch size. We evaluate the benefits of dynamic tree attention in Section 4.4.2 and examine performance improvements through knowledge distillation in Section 4.4.3.

Table 1: Speedup and Tokens/Step on MT-Bench and AlpacaEval.

| MT-bench, temperature=0 | | | | | | |
|---|---|---|---|---|---|---|
| Method | Vicuna 7B | | Vicuna 13B | | Vicuna 33B | |
| | Speedup | Tokens/Step | Speedup | Tokens/Step | Speedup | Tokens/Step |
| Medusa | 2.39x | 2.55 | 2.40x | 2.61 | 2.51x | 2.53 |
| EAGLE | 2.69x | 3.96 | 2.74x | 4.00 | **2.80x** | 3.71 |
| ReDrafter | **2.80x** | **4.20** | **2.80x** | **4.21** | 2.61x | **3.87** |
| MT-bench, temperature=1 | | | | | | |
| Method | Vicuna 7B | | Vicuna 13B | | Vicuna 33B | |
| | Speedup | Tokens/Step | Speedup | Tokens/Step | Speedup | Tokens/Step |
| Medusa | 2.33x | 2.55 | 2.36x | 2.61 | 2.52x | 2.59 |
| EAGLE | 2.31x | 3.51 | 2.27x | 3.55 | 2.63x | 3.48 |
| ReDrafter | **3.50x** | **5.31** | **3.51x** | **5.29** | **3.27x** | **4.69** |
| AlpacaEval, temperature=0 | | | | | | |
| Method | Vicuna 7B | | Vicuna 13B | | Vicuna 33B | |
| | Speedup | Tokens/Step | Speedup | Tokens/Step | Speedup | Tokens/Step |
| Medusa | 2.19x | 2.42 | 2.26x | 2.45 | 2.31x | 2.31 |
| EAGLE | 2.43x | 3.61 | 2.49x | 3.62 | **2.59x** | 3.29 |
| ReDrafter | **2.69x** | **4.06** | **2.78x** | **4.02** | 2.43x | **3.61** |
| AlpacaEval, temperature=1 | | | | | | |
| Method | Vicuna 7B | | Vicuna 13B | | Vicuna 33B | |
| | Speedup | Tokens/Step | Speedup | Tokens/Step | Speedup | Tokens/Step |
| Medusa | 2.26x | 2.42 | 2.24x | 2.50 | 2.34x | 2.43 |
| EAGLE | 2.20x | 3.29 | 2.16x | 3.34 | 2.41x | 3.16 |
| ReDrafter | **3.53x** | **5.31** | **3.50x** | **5.22** | **3.32x** | **4.71** |

## 4.1 PYTORCH BENCHMARK

We compare ReDrafter with Medusa (Cai et al., 2024) and EAGLE (Li et al., 2024a) using PyTorch. For each method, we report speedups relative to auto-regressive decoding, as well as the average number of tokens accepted by LLM per generation step (Tokens/Step) on MT-Bench (Zheng et al., 2024) and Alpaca (Dubois et al., 2024). We conduct experiments at two different temperatures, 0 and 1. Temperature=0 represents the greedy approach (i.e., token matches), while temperature=1 corresponds to rejection sampling.

Table 1 compares different methods. When temperature equals to 1, ReDrafter outperforms Medusa and EAGLE across all LLMs and evaluation datasets, achieving both the highest speedup and highest Tokens/Step. When temperature is 0, ReDrafter attains the highest speedup and Tokens/Step with Vicuna 7B and 13B. For 33B, ReDrafter achieves the highest Tokens/Step, though its speedup is slightly lower than EAGLE's. Notably, ReDrafter achieves approximately 30% greater speedup at temperature 1 compared to temperature 0. This improved performance at temperature 1 can be attributed to the rejection sampling mechanism. At higher temperatures, the distributions of the LLM and draft model become smoother, increasing the likelihood of draft tokens being accepted by rejection sampling.

Figure 5 illustrates that ReDrafter consistently performs well across all model sizes and dataset categories in both MT-Bench and Alpaca. There's a gap between Tokens/Second and speedup, which is anticipated and arises from the overhead associated with the speculative decoding process.

## 4.2 TENSORRT-LLM-BASED CUDA SERVING

To integrate our invention into production-ready environments, we developed a TensoRT-LLM implementation to address high traffic and support long context. We demonstrate significantly improved the CUDA serving performance using Nvidia H100 GPUs.

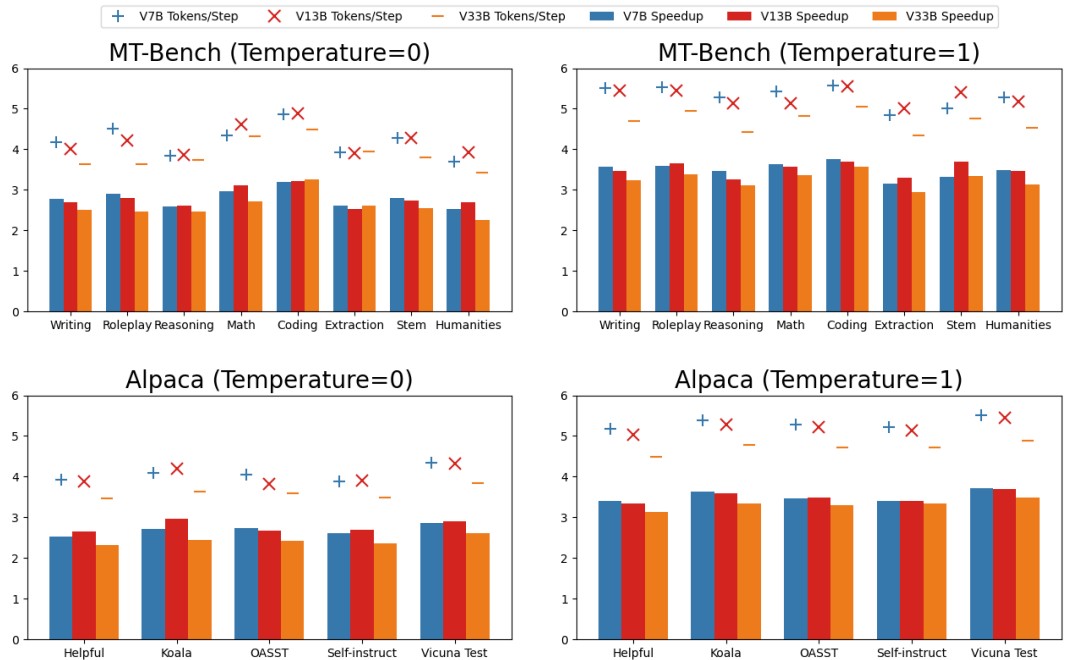

Figure 5: ReDrafter speedup and Tokens/Step on subcategories in MT-Bench and AlpacaEval.

Table 2: TensorRT-LLM experiments on MT-bench with continuous batching at varying request rates, using fixed settings of TP=1, beam width=6, beam length=5, and temperature=1.

| Model | Request Rate Queries Per Second (QPS) | | | | | |
|---|---|---|---|---|---|---|
| | QPS=0.1 | | QPS=0.5 | | QPS=1.0 | |
| | ms/token | Speedup | ms/token | Speedup | ms/token | Speedup |
| V 7B | 2.41 | 2.09x | 2.48 | 2.11x | 2.76 | 1.57x |
| V 13B | 3.41 | 2.31x | 3.61 | 1.71x | 5.43 | 1.22x |
| V 33B | 5.93 | 2.56x | 8.41 | 1.49x | 20.04 | 1.37x |

In production, we run an inference service across multiple GPUs to handle concurrent user requests. To maintain service quality amid fluctuating traffic, latency and throughput are key metrics. Additionally, users typically focus on the latency of the first and last generated tokens. Previous works (Kwon et al., 2023; Kumar et al., 2024; Zhao et al., 2024) introduced techniques such as continuous batching to meet these demands. Given that TensorRT-LLM incorporates these methods, we use our own TensorRT-LLM implementation on 8 GPUs to empirically evaluate the performance improvements achieved with ReDrafter, demonstrated in Table 2.

We observe that ReDrafter significantly enhances the performance across all levels of Queries Per Second (QPS). At QPS 0.1, the speedup ranges from an impressive 2.09x to 2.56x, comparing with the highly optimized auto-regressive baseline in TensorRT-LLM. When QPS raise to 1, the speedup dropped to 1.22x to 1.57x. This is expected, since higher QPS leads to a larger dynamic batch size, pushing the system to a compute limit. We conduct an in-depth investigation of this phenomenon in an experimental environment in Section 4.4.1.

Another requirement for a production system is support for long context. To achieve this, serving systems often shard the model across multiple GPUs, even when it could fit in a single GPU's DRAM, to free up memory for the context. In LLM serving, this sharding is typically done through tensor parallelism (TP, Shoeybi et al. (2019); Karakus et al. (2021); Ko et al. (2021); Gao et al. (2022)). We empirically examine the overhead associated with TP. Table 3 presents TensorRT-LLM Tokens Per Second and Speedup with varying TPs on MT-Bench. We disable continuous batching and set the batch size to 1 to eliminate traffic variability.

Table 3: TensorRT-LLM experiments on MT-bench using static batching with varying tensor parallelism (TP). We fix batch size=1, beam width=6, beam length=5, and temperature=1.

| Model | # Tensor Parallelism (TP) of LLM | | | | | |
| | TP=1 | | TP=2 | | TP=4 | |
| | TPS | Speedup | TPS | Speedup | TPS | Speedup |
|---|---|---|---|---|---|---|
| V 7B | 323.12 | 2.19x | 373.27 | 1.82x | 403.87 | 1.68x |
| V 13B | 198.64 | 2.30x | 251.55 | 1.89x | 312.79 | 1.84x |
| V 33B | 97.50 | 2.51x | 137.47 | 2.15x | 171.11 | 1.93x |

Table 4: MLX experiments with varying beam width (BW). We fix temperature=0, batch size=1, beam length=4. See A.2 for more experimental setup details.

| Chips | Model | BW=1 | | | BW=2 | | | BW=3 | | | BW=4 | | |
| | | TPS | Speedup | Tokens/Step | TPS | Speedup | Tokens/Step | TPS | Speedup | Tokens/Step | TPS | Speedup | Tokens/Step |
|---|---|---|---|---|---|---|---|---|---|---|---|---|---|
| M1 Max | V 7B | **28.22** | **1.32x** | 2.15 | 27.69 | 1.30x | 2.38 | 27.54 | 1.29x | 2.44 | 20.16 | 0.94x | 2.44 |
| M2 Ultra | V 7B | 57.14 | 1.43x | 2.15 | 60.24 | 1.51x | 2.38 | **60.40** | **1.52x** | 2.44 | 30.05 | 0.75x | 2.44 |
| | V 13B | 41.94 | 1.87x | 2.53 | 43.52 | 1.94x | 2.82 | **43.55** | **1.94x** | 2.82 | 21.83 | 0.97x | 2.94 |
| | V 33B | 1.15 | 1.97x | 2.17 | 1.22 | 2.08x | 2.33 | 1.28 | 2.19x | 2.47 | **1.33** | **2.28x** | 2.56 |

For TP=1, ReDrafter accelerates generation by 2.19x to 2.51x, while for TP=4, the speedup ranges from 1.68x to 1.93x. The speedup is less pronounced with higher TPs, when LLM forward pass becomes faster, while the RNN draft model is not fully optimized with TP. The overhead for the draft model constitutes a larger proportion of the overall wall-time, leading to less salient speedup.

Additionally, the observed speedup is not as significant as in the PyTorch experiments (3.27x to 3.50x) in Section 4.1 using the same checkpoint. This discrepancy arises because applying TP to the RNN draft model increases the number of all-gather operations when collecting intermediate tensors from multiple GPUs during beam search, incurring higher overhead compared to PyTorch. Additionally, the AR implementation in TensorRT-LLM is highly optimized, which limits the potential speedup achieved by ReDrafter.

### 4.3 MLX-BASED ON DEVICE INFERENCE

The increasing memory, bandwidth, and computational power of personal devices make it a promising avenue for deploying AI assistants locally. While it's well-known that on-device GPUs have less computational power and bandwidth compared to CUDA-based systems, the on-device scenario is simpler, with a single user interacting with a locally deployed LLM. This setup provides an opportunity to harness available computational resources for speculative decoding.

We benchmarked ReDrafter on Metal GPUs in Apple Silicon chips, specifically the M1 Max and the more powerful M2 Ultra, using an MLX implementation. Table 4 shows promising speedups of 1.37x on M1 Max and higher speedup on M2 Ultra, demonstrating ReDrafter's viability for on-device use case. We omitted experiments for the 13B and 33B models on the M1 Max, as they exceeded the device's memory capacity. In the following, we briefly discuss the experimental results related to beam width, while leaving the implementation details, supplementary results, and other key insights for interested readers in Appendix A.2.

Results in Table 4 show that while higher beam widths consistently yield more tokens per step and reduce the number of LLM calls, the optimal speedup is achieved at lower beam widths. For example, the best performance occurs at BW=1 on the M1 Max for the 7B model, and BW=3 on the M2 Ultra for both the 7B and 13B models. As discussed in Section 3.2, a wider beam requires more floating-point operations (FLOPs) for the LLM to verify the draft tokens. While a powerful GPU can process these FLOPs in parallel, once the computation cost reaches the hardware's limit, the performance gain from speculative decoding diminishes. This explains why the optimal speedup for the M1 Max is achieved at BW=1, whereas for the M2 Ultra, it occurs at BW=3, highlighting the greater computational power of M2 Ultra. However, performance declines at BW=4 for both devices due to the increased cost of verifying draft tokens.

Table 5: Compare per-request tokens/second (TPS), system tokens/second (TPS × BSZ) for Re-Drafter Vicuna 7B with different beam widths (BW) and batch sizes (BSZ) on MT-Bench.

| BSZ | BW=1 | | BW=2 | | BW=4 | | BW=8 | | BW=16 | | BW=32 | | BW=64 | |
|---|---|---|---|---|---|---|---|---|---|---|---|---|---|---|
| | TPS | TPS×BSZ | TPS | TPS×BSZ | TPS | TPS×BSZ | TPS | TPS×BSZ | TPS | TPS×BSZ | TPS | TPS×BSZ | TPS | TPS×BSZ |
| 1 | 62.55 | 62.55 | 73.32 | 73.32 | 80.47 | 80.47 | 88.31 | 88.31 | 100.16 | 100.16 | 104.51 | 104.51 | **110.64** | 110.64 |
| 2 | 58.56 | 117.13 | 70.25 | 140.49 | 75.77 | 151.54 | 86.64 | 173.29 | 99.45 | 198.90 | 107.52 | 215.04 | **111.42** | 222.83 |
| 4 | 57.51 | 230.03 | 67.42 | 269.67 | 74.77 | 299.09 | 81.95 | 327.81 | 97.13 | 388.51 | **99.11** | 396.46 | 85.23 | 340.94 |
| 8 | 53.14 | 425.14 | 60.48 | 483.85 | 69.49 | 555.90 | 81.14 | 649.11 | **83.44** | 667.52 | 73.41 | 587.30 | 54.19 | 433.48 |
| 10 | 53.61 | 536.14 | 58.79 | 587.85 | 66.29 | 662.88 | 71.55 | 715.48 | **73.57** | 735.68 | 65.31 | 653.10 | 44.23 | 442.25 |
| 20 | 45.16 | 903.20 | 49.05 | 980.95 | **57.35** | 1146.95 | 53.87 | 1077.45 | 48.76 | 975.14 | 35.66 | 713.17 | 23.28 | 465.70 |
| 40 | 32.69 | 1307.65 | 33.59 | 1343.41 | **36.68** | 1467.04 | 34.33 | 1373.08 | 26.43 | 1057.20 | 19.15 | 766.03 | 12.09 | 483.58 |
| 80 | 18.94 | 1515.40 | **20.45** | 1636.36 | 19.65 | 1571.89 | OOM | OOM | OOM | OOM | OOM | OOM | OOM | OOM |

We observed the TPS drops sharply from 43.55 to 1.33 when the model size increases from 13B to 33B, yet a significant speedup is still achieved. Our conjecture is that the 33B model encounters a memory/IO bottleneck due to the increased model size, causing inference to be dominated by memory swapping and data transfer, which significantly reduces TPS. With the investigation above, we believe ReDrafter holds significant potential for further improvement as on-device hardware continues to evolve. However, for larger models, compression techniques like quantization may be necessary to achieve acceptable latency.

### 4.4 ABLATION STUDY IN PYTORCH

#### 4.4.1 BEAM WIDTH AND BATCH SIZE

In ReDrafter, beam width and batch size are two critical factors for leveraging redundant computational resources. While increasing either may push the system to its computational limits, they impact the system differently: a larger beam width raises the likelihood of generating draft sequences that are more likely to be accepted, whereas a larger batch size improves overall system throughput.

We examine the trade-offs between beam width and batch size by conducting a grid search on both parameters on an Nvidia H100 GPU using MT-Bench, with Vicuna-7B as the LLM. We measured tokens-per-second per request (TPS) to assess latency (the inverse of latency) and tokens-per-second per request multiplied by batch size (TPS×BSZ) to evaluate overall system throughput. The results are summarized in Table 5.

When beam width is held constant and batch size increases, TPS initially remains stable but eventually declines, while TPS×BSZ continues to rise. This shows that increasing batch size improves GPU utilization and enables more efficient batch processing, despite fewer resources being allocated to each individual request. The TPS drop occurs earlier at higher beam widths due to the additional computational cost. At a batch size of 80, out-of-memory (OOM) errors occur at larger beam widths, indicating limited memory capacity.

The optimal configurations for per-request TPS and TPS × BSZ vary. The highest per-request TPS, around 110, is achieved with a beam width of 64 and a batch size of 1 or 2, while the highest TPS×BSZ, approximately 1636, is reached with a beam width of 2 and a batch size of 80. This underscores the importance of tuning beam width and batch size based on specific use cases. For scenarios prioritizing low latency, a larger beam width with a smaller batch size is recommended. Conversely, if high throughput is the main goal, a larger batch size paired with a moderate beam width is more effective.

#### 4.4.2 DYNAMIC TREE ATTENTION

As outlined in Section 3.3, dynamic tree attention significantly reducing computational cost by de-duplicating draft tokens. We study its effectiveness and demonstrate empirical results in Figure 6.

The computational gain from using train attention is determined by the compression ratio, which is the number of tokens in the beam divided by the number of tokens in the packed beam. We conducted an empirical study on MT-Bench, using Vicuna 7B as the LLM, with a fixed batch size of 1, beam length of 5 and varying the beam width from 5 to 70. This results in token counts per beam ranging from 25 to 350. We then measured the average number of tokens in the packed beam after applying dynamic tree attention. As shown in Figure 6 (left), dynamic tree attention effectively

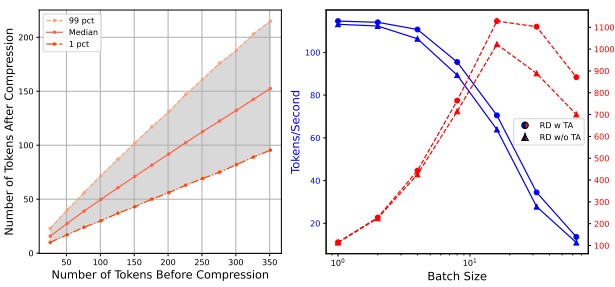

Figure 6: Left: Number of tokens to be verified by LLM before and after the compression using dynamic tree attention. Right: Per-request tokens per second and overall tokens per second for ReDrafter w, w/o dynamic tree attention. RD is an abbreviation for ReDrafter.

reduces the number of draft tokens by 30% to 60%. It also demonstrates a consistent compression ratio across different beam sizes, with the ratio remaining predictable even in extreme cases (99th and 1st percentiles).

We demonstrate that dynamic tree attention can further enhance performance under computational constraints. We fix beam length to 5, beam width to 45, and tune batch size to push the compute limit, Figure 6 (right) shows that when the batch size is below 4, computational resources are abundant, and there is no significant difference in TPS or TPS×BSZ. However, when the batch size exceeds 4, we encounter a computational bottleneck. In this scenario, ReDrafter with dynamic tree attention (RD w TA) significantly outperforms ReDrafter without tree attention, delivering higher throughput and more tokens per second. In practical deployment, both speed and throughput should be considered to make balanced decisions.

### 4.4.3 KNOWLEDGE DISTILLATION

Table 6: Compare Speedup and Average Accepted Tokens Per Step (Tokens/Step) for ReDrafter using Vicuna 7B with and w/o distillation. Batch size is 1.

| Distillation | BW=1 | | BW=2 | | BW=4 | | BW=16 | | BW=64 | |
|---|---|---|---|---|---|---|---|---|---|---|
| | Speedup | Tokens/Step | Speedup | Tokens/Step | Speedup | Tokens/Step | Speedup | Tokens/Step | Speedup | Tokens/Step |
| N | 1.47 | 2.21 | 1.52 | 2.31 | 1.54 | 2.48 | 1.80 | 2.87 | 1.99 | 3.30 |
| Y | 1.54 | 2.35 | 1.60 | 2.50 | 1.72 | 2.73 | 1.92 | 3.09 | 2.18 | 3.58 |

As discussed in Section 3.5, a more effective objective for training draft models is to align with LLM predictions through knowledge distillation, rather than simply fitting ground-truth tokens. To test this hypothesis, we trained one draft model using a distilled dataset and another using ground-truth tokens, both based on Vicuna 7B. The distilled dataset was created by having the LLM generates 5 future tokens at each position of the ground-truth response using a temperature of 0. Table 6 shows that distillation lead to an approximate 10% increase in the speedup and the average accepted tokens per step. This demonstrates that training with distillation offers a tangible performance boost, improving both generation efficiency and predictive accuracy.

## 5 FUTURE WORK

ReDrafter sets the stage for pushing the speedup limits of speculative decoding across various hardware and implementations through its novel design, utilizing an RNN draft model along with tailored training and inference algorithms. While achieving state-of-the-art performance, we identify areas for further improvement, such as enhancing draft model training through more advanced distillation techniques, and optimizing implementation to ensure consistent performance gains and less overhead. We leave those to future work.

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

# A APPENDIX

## A.1 GPU-FRIENDLY DYNAMIC TREE ATTENTION IMPLEMENTATION

In Listing 1, we demonstrate the GPU-friendly implementation of the function `dedup_prefix` from Section 3.3 with five lines of code. We assume batch size equals to 1 in this example for brevity. Extending to batch size greater than 1 is straight-forward by pre-pending an extra batch size dimension in tensors.

The function processes the initial tensor `beam`, generating the `prefix_tree` tensor as its output. In this tensor, `prefix_tree[i][j]=k` indicates that the candidate at the smallest index $k$ shares an identical prefix `beam[i][:j+1]`. For instance, `prefix_tree[2][1]=0` signifies a shared prefix "*morning sipping*" between `beam[0]` and `beam[2]` from the example in Figure 4. It is possible to condense tokens where `prefix_tree[i][j]<i`.

Listing 1: An example inplementation for dedup_prefix.

```python
def dedup_prefix(beam):
    """For each prefix in each candidate, find the smallest candidate
        index that shares the same prefix.

    Args:
        - beam: (beam_width, beam_length) input beam.
    Returns:
        - prefix_tree: [beam_width, beam_length] prefix_tree[i][j]=k
            indicates that candidate sequences i and k in share the same
            prefix, or, in other words, beam[i][:j+1]== beam[k][:j+1]
    Examples:
        beam = tensor([[91, 92, 93, 95],
                       [91, 92, 94, 96],
                       [91, 92, 93, 97]])
        prefix_tree = tensor([[0, 0, 0, 0],
                              [0, 0, 1, 1],
                              [0, 0, 0, 2]])
    """
    beam_length = beam.shape[1]
    prefix_target = torch.arange(1, beam_length+1)
    # Build a square boolean matrix matches.
    # If matches[i][j][k]==True, then the k-th token of the i-th sequence
    #     is the same as the k-th token in the j-th sequence.
    # So, i and j are in range [0,beam_width), k in [0,beam_length).
    matches = beam[:, :, None]==beam[:, None, :]
    seq_matches = (torch.cumsum(matches, dim=2)
        == prefix_target[None, None, :])
    # The previous candidate with smallest index that shares the same
    #     prefix.
    prefix_tree = torch.argmax(seq_matches, dim=2)
    return prefix_tree
```

## A.2 BENCHMARK RECURRENT DRAFTING, A FAST SPECULATIVE DECODING METHOD, IN MLX

### A.2.1 OPTIMAL SPEEDUP AND GPU POTENTIAL

To gain detailed insights into the behavior of Recurrent Drafting on Apple Silicon chips, we need to analyze the speedup in relation to two key inference-time parameters: beam width and beam length in the drafter beam search.

The parameter beam width refers to the number of draft token sequences. A larger beam width increases the likelihood that the sequence with the longest acceptable prefix is included in the beam. This allows the LLM to accept more tokens in each decoding step, reducing the overall number of decoding steps required—or, equivalently, decreasing the number of calls to the LLM.

However, a wider beam requires more FLOPs for the LLM to verify the draft tokens. While a powerful GPU can process these FLOPs in parallel, minimizing the increase in wall time, it also implies higher power consumption and computational cost.

The parameter beam length refers to the number of steps in the beam search algorithm, the number of calls to the RNN draft model, or equivalently, the length of the draft token sequences in the beam. A longer beam increases the length of the prefix that can be accepted, reducing the number of decoding steps or calls to the LLM. However, this length is constrained by the token sequence length used during the training of the RNN draft model.

Figure 7 illustrates the speedup in relation to beam width and beam length on the M1 Max and M2 Ultra. We can observe that optimal speedup (colored in yellow) is achieved when the beam length is close to the length of the training sequences used for the RNN draft model. This maximizes the predictive power of the RNN. However, the optimal speedup is not necessarily achieved at the exact training length of 5; it can occur at a slightly shorter length, such as 4, since the RNN may not always accurately predict the 5th token.

We also observe that the optimal speedup is achieved with narrower beams—1 for the M1 Max and 3 for the M2 Ultra. In contrast, our experiments on A100 and H100 GPUs using PyTorch show that the optimal speedup is achieved with beam widths of 50 or more. This discrepancy is due to the inherent performance gap between server-grade GPUs and mobile GPUs, which is expected. It also explains why the M2 Ultra can handle a wider optimal beam of 3, compared to the M1 Max's 1, as the M2 Ultra is equipped with a more powerful GPU.

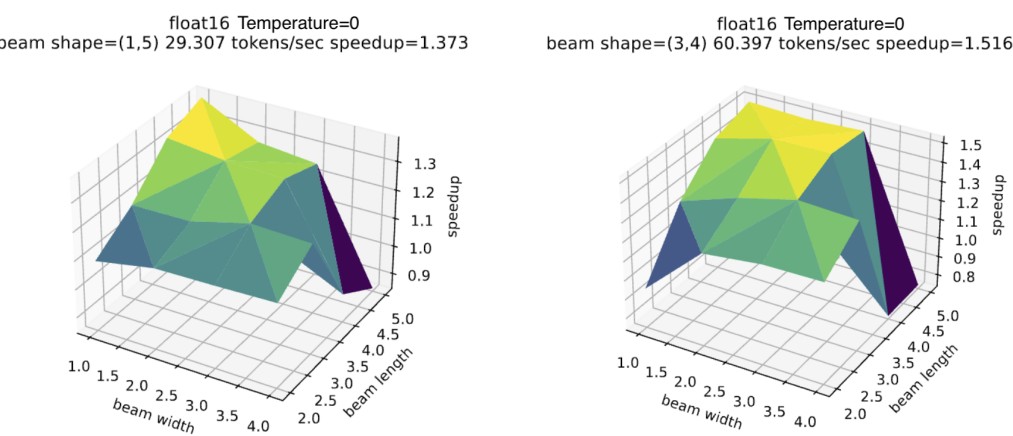

Figure 7: Tokens Per Second and Speedup of ReDrafter with Vicuna 7B on M1 Max and M2 Ultra.

### A.2.2 PERFORMANCE TUNING: MLX AND APPLE SILICON

Before starting the MLX implementation, we had experience with PyTorch and TensorRT-LLM. However, working on our first MLX project revealed that many lessons we learned from programming CUDA no longer apply to MLX and Apple Silicon. Below, we share some of the key lessons learned from this journey.

**Use Native Dtype and Low-Bits**

Our benchmark code explores various factors, including data types (dtype). From our exploration, running autoregression and recurrent drafting in float16 is consistently faster than in `bfloat16`. We also observed that both `float16` and `bfloat16` outperform `float32`, as they use less memory access bandwidth. Similarly, as reported by other studies, 4-bit quantization significantly accelerates MLX programs compared to `float16`.

**MLX Does Lazy Evaluation**

While PyTorch programs execute eagerly, MLX programs run lazily. In MLX, the Python code may appear to be executing tensor operations sequentially, but these operations might not actually run until the final result is printed. This lazy execution extends beyond numerical operations; even

loading model parameters from the filesystem into tensors in memory is done lazily. To ensure that a model is fully loaded before benchmarking the inference algorithm, you must call `mlx.core.eval(model.parameters())` before invoking the model.

**Don't Break the JIT Compilation**

Lazy evaluation in MLX allows it to silently trace tensor operation calls and compile these traces just-in-time into Metal GPU kernels for improved runtime performance. While this provides convenience, it can also influence how we program. For example, during a comparison of the execution time between MLX functions and their PyTorch counterparts, we noticed that one MLX function consumed a disproportionately large fraction of the total running time, whereas its PyTorch equivalent did not. We discovered that this function involved numerous calls to `array.item()`. MLX had to compile the Python code before each of these calls, causing the code to be split into many segments, which significantly slowed down the execution time.

**Measure Performance in All Levels of Details**

The Instruments app, included with Xcode, provides visualization of CPU threads and Metal GPU operations, similar to NVIDIA Nsight. This tool helps developers gain a general understanding of potential bottlenecks in their code. to capture detailed performance metrics, we developed custom utilities to measure and log the execution time of individual functions.

