# OpenReview forum: "Recurrent Drafter for Fast Speculative Decoding in Large Language Models"
_ICLR.cc/2025/Conference — Submitted to ICLR 2025_

### Official Review · Reviewer_Uws2 · 2024-10-29

**Soundness:** 2
**Presentation:** 3
**Contribution:** 2
**Rating:** 6
**Confidence:** 3

**Summary:**

This paper introduces ReDrafter, which achieves state-of-the-art speed improvements for LLM inference in speculative decoding through the use of RNNs, dynamic tree attention, and knowledge distillation from the LLM. ReDrafter demonstrates significant speed enhancements compared to baseline methods across various frameworks and chips.

**Strengths:**

1. This paper tackles a significant issue in LLM inference by promoting speculative decoding, demonstrating considerable speed improvements compared to state-of-the-art methods like Medusa and Eagle.
2. The method has been extensively tested across various chips and frameworks, including H100, Apple’s M2, TensorRT-LLM, MLX, and PyTorch.
3. The paper is well written and effectively conveys the main design of the algorithm.

**Weaknesses:**

1. Could you include a comparison with Eagle-2? It's listed in your references but isn’t included in the evaluation.

**Questions:**

See weakness. Thanks!

---

> ### Author Response · Authors · 2024-11-25
> **Response**
>
> Thanks for your comments.
>
> > Compare with Eagle2.
>
> We located Eagle2's [1] the source code and checkpoint on the authors' website and successfully reproduced their results on MT-Bench and Alpaca within our experimental environment with a single H100 GPU. This approach ensures consistency with ReDrafter's setup. The reproduced results are presented below.
>
> **_Inference Speed-up VS. AR ↑\| Tokens/Step↑_**
> Model \| Dataset|Mt-Bench,t=0|Alpaca,t=0|Mt-Bench,t=1|Alpaca,t=1
> -|-|-|-|-|
> EAGLE2-V-7B|**3.00x \| 4.63**|**2.80x \| 4.28**|2.74x \| 4.10|2.66x \| 3.88
> ReDrafter-V-7B|2.80x \| 4.20|2.69x \| 4.06|**3.50x \| 5.31**|**3.53x \| 5.31**
> |
> EAGLE2-V-13B|**3.13x \| 4.61**|**3.18x \| 4.38**|2.86x \| 4.18|2.74x \| 3.97
> ReDrafter-V-13B|2.80x \| 4.21|2.78x \| 4.02|**3.51x \| 5.29**|**3.50x \| 5.22**
> |
>
> In the results presented above, ReDrafter surpasses EAGLE2 at a temperature setting of 1, highlighting the advantage of beam search in managing uncertainty within the distribution. On the other hand, EAGLE2 shows superior performance at a temperature of 0, owing to its use of a greedy search approach for token tree construction and additional optimizations tailored for zero-temperature conditions.
>
> [1] Y. Li et al. EAGLE-2: Faster Inference of Language Models with Dynamic Draft Trees.

---

> > ### Comment · Reviewer_Uws2 · 2024-11-26
> >
> > Thanks for your new results. I will raise my score to 6.

---

> > > ### Author Response · Authors · 2024-11-28
> > >
> > > We’re pleased to know that your concerns have been resolved. Thank you for increasing your score.

---

### Official Review · Reviewer_SwoJ · 2024-10-29

**Soundness:** 3
**Presentation:** 3
**Contribution:** 3
**Rating:** 6
**Confidence:** 3

**Summary:**

This paper aims to accelerate the inference of LLMs. The proposed architecture utilized speculative decoding technology and applied a draft model with an RNN structure. They also use dynamic tree attention to further accelerate inference.

**Strengths:**

1.The paper is well-written and flows very smoothly. The framework and workflow of the entire method are clearly articulated.
2.This paper conduct extensive experiments to demonstrate the effectiveness of their approach, covering various models and different devices.

**Weaknesses:**

1. Although there is textual description of ReDrafter, providing a structural diagram would better assist readers in understanding the structure of the ReDrafter Model.
2. I believe the paper lacks sufficient explanation for the idea of using an RNN-based draft model. For example, I think it would be beneficial to include an experiment that substitutes the RNN with a single-layer decoder model to observe the resulting changes.
3.Combining embeddings and hidden states as input is not a novel approach.[1]
4. The article mentions that tree attention is dynamic and states that it "relies on individual beam search results," but it does not explicitly explain the relationship. Alternatively, what does the author consider to be “static” tree attention? If it does indeed exist, I believe it should be compared in the ablation experiment section between dynamic TA, static TA, and without TA.

[1] EAGLE: Speculative Sampling Requires Rethinking Feature Uncertainty, ICML 2024

**Questions:**

1. Although there is textual description of ReDrafter, providing a structural diagram would better assist readers in understanding the structure of the ReDrafter Model.
2. I believe the paper lacks sufficient explanation for the idea of using an RNN-based draft model. For example, I think it would be beneficial to include an experiment that substitutes the RNN with a single-layer decoder model to observe the resulting changes.
3.Combining embeddings and hidden states as input is not a novel approach.[1]
4. The article mentions that tree attention is dynamic and states that it "relies on individual beam search results," but it does not explicitly explain the relationship. Alternatively, what does the author consider to be “static” tree attention? If it does indeed exist, I believe it should be compared in the ablation experiment section between dynamic TA, static TA, and without TA.

[1] EAGLE: Speculative Sampling Requires Rethinking Feature Uncertainty, ICML 2024

---

> ### Author Response · Authors · 2024-11-25
> **Response**
>
> Thanks for your comments.
>
> To make it more clear for readers, we begin by presenting a structured workflow for ReDrafter:
>
> ---
> 1. Initialize - Start with the previously generated tokens and the last hidden state.
>
> 2. **While** stopping criteria are not met:
>    2.1. Draft Model Generation - Use beam search to generate a beam (set of candidate sequences).
>    2.2. Dynamic Tree Attention - Flatten and compress the beam into a packed beam. Then formulate an attention mask.
>    2.3. LLM Verification - Perform a forward pass through the LLM to compute log probabilities for all proposed tokens.
>    2.4. Candidate Selection - Select the best candidate with the longest accepted prefix.
>    2.5. Update State - Append accepted tokens to the prefix and update hidden states.
> 3. **Return**  The final generated sequence.
> ---
>
> Next, we present additional ablation results to address your questions.
>
> > Substitutes the RNN with a single-layer decoder.
>
> Thank you for the suggestion. We have included this analysis and provided the results to assess the impact of the RNN component on ReDrafter's performance on MT-Bench when temperature is zero. Our findings show that the RNN is more effective at accurately predicting the next few tokens compared to a one-layer decoder:
>
> **_Inference Speed-up VS. AR ↑ \| Tokens/Step ↑_**
>
> Model \| Draft Model Arch. | RNN | One-Layer Decoder
> -|-|-|
> ReDrafter-V-7B| **2.80x \| 4.20** | 2.06x \| 3.09
> ReDrafter-V-13B|**2.80x \| 4.21**|  2.21x \| 3.43
>
> > Compare dynamic TA, static TA, and without TA.
>
> By "static tree attention," we refer to the draft model speculating a token tree without context or dependency consideration. Both Medusa and EAGLE use this approach, selecting high-probability tree tokens offline instead of using beam search during runtime. Our comparison with them highlights the benefits of dynamic tree attention, but we're happy to provide additional results by implementing static tree attention in ReDrafter.
>
> **_Inference Speed-up VS. AR ↑ \| Tokens/Step ↑_**
>
> Model \| Method | Dynamic TA | Static TA
> -|-|-|
> ReDrafter-V-7B| **2.80x \| 4.20** | 2.11 \| 3.29
> ReDrafter-V-13B| **2.80x \| 4.21** | 1.92 \| 3.04
>
> In the paper, we compare experimental results with and without tree attention (see Figure 6). Tree attention reduces memory and compute usage by pruning duplicate tokens, enabling a larger number of requests to be served with limited resources. It does not affect tokens per step. The best way to evaluate the difference is by measuring throughput. In our experimental setup, we use batch_size * tokens/second as a proxy for throughput.
>
> **_Max Batch Size x Tokens/Step ↑_**
>
> Model \| Method | Dynamic TA | No Dynamic TA
> -|-|-|
> ReDrafter-V-7B| **1114.48** | 1013.55

---

### Official Review · Reviewer_EUGA · 2024-11-03

**Soundness:** 3
**Presentation:** 3
**Contribution:** 3
**Rating:** 5
**Confidence:** 3

**Summary:**

The paper proposes a new speculative decoding algorithm (ReDrafter), that uses the final hidden state of the LLM as input to an RNN. The speedup is further enhanced by using dynamic tree attention over beam search results for the draft model. ReDrafter provides speedups over other decoding algorithms such as EAGLE and Medusa.  The algorithm is implemented and compared with other decoding techniques in multiple frameworks (Pytorch, TensorRT-LLM,  MLX on Apple M2 Ultra Metal GPU). Finally, a new training objective for the draft model is proposed based on knowledge distillation to better align the draft model with the target LLM.

**Strengths:**

Several ablation studies are done to show the effectiveness of all the main contributions, including ablations for the knowledge distillation training objective and the ablation for the
dynamic tree attention. This makes the contributions more clear.

The algorithm is implemented in several frameworks, and performance improvements are shown in all frameworks, making the practical significance of the algorithm higher.

The work is quite original while building on previous work on knowledge distillation and using tree structures to save computation.

The ReDrafter approach gives significant speedups over EAGLE and Medusa on several frameworks and model sizes.

**Weaknesses:**

The paper does not seem to include details on the compute resource/time requirements to train the draft model. This is important to understand for others wanting to train ReDrafter on their local LLM.

For many LLMs, there exist smaller LLMs from the same family that can be used as draft models. It would have been good to compare against approaches using separate draft models.

Very minor comments:
There is a parenthesis missing in equation 1.
In the first sentence of the abstract, I would use LLM rather than LLMs.

**Questions:**

Will the code be open sourced?

Can you clarify why you are only comparing against the EAGLE and Medusa papers?

How is the performance of ReDrafter compared to approaches that use a separate draft model?

---

> ### Author Response · Authors · 2024-11-25
> **Response**
>
> Thanks for your comments.
>
> > Compute resource/time requirements to train the draft model.
>
> There are two steps to train a draft model: (i) training (ii) (optional) data distillation. We analyze their cost separately.
>
> As our draft model is attached to the LLM, training can be done more efficiently conditioning on LLM's hidden states. In that case, we can skip pretraining stage of the draft model and directly start with SFT without losing accuracy. The time cost of training draft models on ShareGPT dataset for 2 epochs based on Vicuna 7B, 13B, 33B with 8 H100 GPUs is as follows. In practice, ReDrafter-V-7B converges after training for 2 epochs. ReDrafter for larger draft model requires less training epochs.
>
> We perform data distillation as an optional preliminary step before model training. The ShareGPT dataset is divided into 121 shards and processed using 8 H100 GPUs. The costs are outlined below. Since data distillation for each LLM is a one-time offline process, users can weigh the benefits (see Table 6 in the paper) against the costs to decide whether to use it.
>
> Model \| Cost| Training Time | Offline Distillation Time
> -|-|-|
> ReDrafter-V-7B|1.5hr|12hr
> ReDrafter-V-13B|2hr|20hr
> ReDrafter-V-33B|4hr|45hr
>
> > Smaller LLMs as standalone draft models.
>
> We used Vicuna-7B as the draft model for Vicuna-33B and compare with ReDrafter's result as follows. ReDrafter with Vicuna-33B achieves higher accepted tokens per step while using only 1B parameters, highlighting its superior parameter efficiency.
>
> **_Accept Tokens Per Step ↑_**
> LLM \| Draft Model|Vicuna-7B |ReDrafter
> -|-|-|
> Vicuna-33B|3.71| 3.87
> |
>
> > Open source code.
>
> We commit to open-sourcing our code upon acceptance. In the meantime, we provide a preliminary zipped code package (see supplementary material) to help reproduce our results while maintaining anonymity.
>
> > Why only comparing against EAGLE and Medusa?
>
> We recognize the broad range of research related to ReDrafter. For our comparisons, we chose EAGLE and Medusa as both utilize a draft-head-attached-to-LLM framework, allowing us to highlight ReDrafter's key strengths: (i) superior draft model design (compared to Medusa) and (ii) dynamic tree attention (compared to EAGLE). Furthermore, we extended our comparisons to include Sequoia and EAGLE2. ReDrafter achieves significant speedups over Sequoia on Vicuna-33B (2.61x vs. 1.82x) using the MT-Bench dataset and outperforms EAGLE2 in speedup performance at a temperature of 1 on Vicuna-7B (3.50x vs. 2.74x) and 13B (3.51x vs. 2.86x). For additional details, please refer to our responses to Reviewers zZts and Uws2.

---

> > ### Comment · Reviewer_EUGA · 2024-12-02
> > **Official Comment by Reviewer EUGA**
> >
> > Thank you for your response, these have addressed my concerns in the initial review. However, I think that the claim that the output distriubtion of the algorithm is maintained is important, and upon more careful review, it's also unclear to me (see the response from zZts) if the output distribution is maintained using ReDrafter.
> >
> > Due to this, I will lower my score to a 5, and I hope the authors will clarify.

---

> > > ### Author Response · Authors · 2024-12-04
> > >
> > > Thank you for your thoughtful reply. As we explained in detail in our response to Reviewer zZts, beam search, being a deterministic greedy search algorithm, generates candidate sequences that do not strictly adhere to the distribution defined by the draft model. This means it does not guarantee distribution matching when using speculative decoding. To address this, we proposed two fixes in the response, and with these adjustments, ReDrafter continues to deliver the best speed-up performance compared to Medusa and EAGLE. We will ensure these details are included in the paper.

---

### Official Review · Reviewer_zZts · 2024-11-05

**Soundness:** 2
**Presentation:** 3
**Contribution:** 2
**Rating:** 3
**Confidence:** 4

**Summary:**

This paper proposes "ReDrafter", a speculative decoding algorithm that uses an RNN as the draft model. The RNN takes as input the outputs of the LLMs last hidden layer, and then uses beam search to construct a set of possible continuations, which are then verified using an efficient "dynamic tree attention" algorithm by the target model. The RNN is trained using knowledge distillation, to maximize alignment with the target model. Empirically, ReDrafter shows large speedups (on both H100s in PyTorch/TensorRT-LLM, and on Apple Silicon in MLX), generally attaining meaningfully larger speedups than Eagle and Medusa.

**Update**: After reviewing the authors’ responses, I have lowered my score to a 3, as I remain unconvinced that beam search can be used by the draft model during stochastic decoding while maintaining the output distribution of the target model.

**Strengths:**

- Using an RNN as a draft model, taking as input the outputs of the target model's last hidden layer, is a nice idea. This allows introducing dependence between the draft model's speculated tokens (unlike Medusa), while also leveraging the strong representations from the target model (like Eagle), while not requiring storing a KV cache for the draft model.
- The empirical speedups attained by ReDrafter appear quite large.

**Weaknesses:**

- **Most importantly**: It's unclear to me how beam search can be used by the draft model while maintaining the guarantee that the output distribution of the target model is unchanged, when stochastic decoding is used. Can you please clarify?
- It seems that a large percentage of the ReDrafter speedups may come from the large beams used during decoding, as opposed to from an improved draft model architecture. I would have appreciated more careful ablations to clarify this. For example, could one use a transformer draft model (e.g., Llama-3.2-1B for Llama-3.1-70B) with beam search, and attain similar/larger speedups than the RNN draft model? Could you show the relative speeds and acceptance rates of ReDrafter vs. standalone draft models / Medusa / Eagle, to understand the relative merits of these different draft model architectures? How do these different draft model architectures compare when a simple chain of tokens is speculated (like in Leviathan et al), instead of a tree of tokens?
- Given that much of the speedups seem to come from the beam search process, it's important to compare with methods like Sequoia and Eagle2, which also perform speculative decoding using large trees of tokens (instead of simply token sequences).
- It would have been nice to see experiments with Llama 3 models.

**Questions:**

- In the equation in line 156, it doesn't seem $h$ or $g_t$ are used in the recurrence equation for $s_t$. Can you please clarify?

---

> ### Author Response · Authors · 2024-11-25
> **Response (1/2)**
>
> Thanks for your comments.
>
> Below, we use an example to thoroughly explain the theoretical guarantee of ReDrafter in producing the same distribution as the LLM. Regarding your question:
>
> > Why is the output distribution unchanged?
>
> This is guaranteed through the verification stage in speculative decoding, either with greedy decoding (when t=0) or speculative sampling (when t>0).
>
> - A greedy decode example: if the drafter proposes `I like tea` and the LLM outputs `I am bob`, only the first token I is accepted to ensure a match.
>
> - Speculative sampling: When we sample tokens from the drafter, the beam search method returns their log probabilities log Q(v), in addition to v.  To verify v, we pass the token before v to the LLM, which returns log P(v).  Given v, log Q(v), and log P(v), we follow the routine:
>   - Step 1: accept v with probability min(1,P(v)/Q(v)).
>   - Step 2.1: if v is accepted, continue to the next token proposed by the drafter.
>   - Step 2.2: if v is rejected, sample v from an un-normalized distribution R(v)=max(0, P(v)-Q(v)).
>
>   Consider a simple vocabulary with only two tokens: A and B.  Assume that Q(A)=Q(B)=1/2, P(A)=1/4, and P(B)=3/4.
>
>   ```
>                                 P(B)=3/4
>                                 ▄▄
>   Q(A)=1/2             Q(B)=1/2 ██
>        ┌─┐ P(A)=1/4         ┌─┐ ██
>        │ │ ▄▄               │ │ ██
>        └─┘ ▀▀               └─┘ ▀▀
>   ----------------------------------------
>         A                     B
>   ```
>
>   Assume that sampling from Q(v) produces a token A. Since the true probability of A, P(A), is lower than Q(A), we accept A with a probability of P(A)/Q(A)=1/2​ and reject it with a probability of 1-P(A)/Q(A)=1/2. Now, assume that sampling from  Q(v) produces a token B. Since the true probability P(B) is higher than Q(B), we always accept B. This procedure results in the following outcomes:
>   - Acceptance of v=A with probability 1/4.
>   - Acceptance of v=B with probability 1/2.
>   - Rejection with probability 1/4.
>
>   If the token is rejected, we resample from a tilted, unnormalized distribution R(v)=max(0,P(v)−Q(v)).
>
>   ```
>                     R(B)=1/4
>                     ▄▄
>         R(A)=0      ▀▀
>   -------------------------
>         A           B
>   ```
>
>   In this case, R(A)=0, R(B)=1/4, meaning sampling from R yields B almost surely. The overall probably of selecting B is 1/2+1/4=3/4, which matches the LLM's probability.
>
>   Append A.1 of the paper at [1] shows that the above intuition is exactly what we need to do to alter drafter samples such that they look like LLM samples.
>
> [1] Y. Leviathan et al. Fast Inference from Transformers via Speculative Decoding. ICML 2023.

---

> ### Author Response · Authors · 2024-11-25
> **Response (2/2)**
>
> Below, we present additional ablation studies on ReDrafter. We compare it against standalone methods to highlight the effectiveness of incorporating attached draft models. Additionally, we demonstrate the advantages of utilizing a tree structure for speculative tokens over a linear chain. We also include a comparison with recently proposed speculative decoding methods.
>
> > ReDrafter vs. standalone draft models.
>
> We appreciate your interest in comparing draft models attached to LLMs with standalone draft models. Since the training data for Llama 3.2 is not open-sourced, we were unable to train a ReDrafter for comparison. Instead, we used Vicuna-7B as the draft model for Vicuna-33B, achieving **3.71** accepted tokens per step. In comparison, ReDrafter with Vicuna-33B achieves **3.87** accepted tokens per step while using only 1B parameters, highlighting its superior parameter efficiency.
>
> We must note that using standalone draft models results in restrictions and cost that cannot be accounted for in the quantitative results, such as:
> 1. A standalone draft model must share the same tokenizer as the LLM.
> 2. Standalone draft models often require intricate alignment of pre-filling and KV-cache, complicating deployment.
>
> ReDrafter avoids these issues, ensuring broader compatibility. Also, ReDrafter can quickly adapt to the LLM during training. It does not need to train from scratch. In our experiments we only train the ReDrafter on ShareGPT for a few epochs, which only cost less than 5 hours on 8 H100 GPUs.
>
> > Using a chain of speculative tokens.
>
> When ReDrafter operates with a beam width of 1, it simplifies to a chain of speculative tokens. In this case, ReDrafter-V-7B achieves a 1.79x speedup on MT-Bench at t=0, compared to the 2.80x speedup observed with beam search. This highlights the effectiveness of allocating additional compute during inference time through beam search.
>
> > Compare with Sequoia and Eagle2.
>
> Sequoia [2] and Eagle2 [3] authors provide their codebase and checkpoints in their repositories, allowing us to reproduce some of their results. We made adjustments to align the code with ReDrafter's experimental setup, as detailed below.
>
> - **Sequoia experiments**: Sequoia reports results on Vicuna 33B. Utilizing the authors' codebase, we adapted it to the Mt-Bench dataset and executed their implementation on a single H100 GPU in offloading mode. Without offloading, we encountered an out-of-memory (OOM) issue.
>
>     **_Inference Speed-up VS. AR ↑ \| Tokens/Step ↑_**
>     Model \| Temperature|t=0|t=1
>     -|-|-|
>     Sequoia-V-33B|1.82x \| 1.84 |1.45x \| 1.46
>     ReDrafter-V-33B|**2.61x \| 3.87**|**3.27x \| 4.69**
>     |
>
> - **EAGLE2 experiments**: EAGLE2 authors reports results on Vicuna 7B and 13B in their preliminary draft. Below, we reproduce these results in our experimental setup on Mt-Bench using a single H100 GPU.
>
>     **_Inference Speed-up VS. AR ↑ \| Tokens/Step ↑_**
>     Model \| Temperature|t=0|t=1
>     -|-|-|
>     EAGLE2-V-7B|**3.00x \| 4.63**|2.74x \| 4.10
>     ReDrafter-V-7B|2.80x \| 4.20|**3.50x \| 5.31**
>     |
>     EAGLE2-V-13B|**3.13x \| 4.61**|2.86x \| 4.18
>     ReDrafter-V-13B|2.80x \| 4.21|**3.51x \| 5.29**
>     |
>
>     In the results above, ReDrafter outperforms EAGLE2 when the temperature is set to 1, demonstrating the effectiveness of beam search in handling uncertainty within the distribution. Conversely, EAGLE2 performs better when the temperature is set to 0. This is because EAGLE2 employs a greedy search strategy to construct the token tree and incorporates additional acceleration optimizations specifically for zero-temperature scenarios.
>
> > Equation in line 156.
>
> We concatenate $h$ on $s_t$ as $g_t$ to forecast token at position $t$. This is a special case of RNN with $h_t=h$, that is, an identity transition matrix on $h$. This could help enhance longer range forcasting and get rid of short-sighted errors. See [4] for more intuitions.
>
> [2] Z. Chen et al. Sequoia: Scalable and Robust Speculative Decoding. NeurIPS 2024.\
> [3] Y. Li et al. EAGLE-2: Faster Inference of Language Models with Dynamic Draft Trees.\
> [4] G. Bachmann and V. Nagarajan. The Pitfalls of Next-Token Prediction. ICML 2024.

---

> ### Comment · Reviewer_zZts · 2024-12-02
>
> I thank the authors for their thoughtful responses.
>
> I remain concerned about the correctness of the proposed algorithm, during stochastic decoding. My concern is that in beam search, even if we know the true probabilities (according to the draft model) of all of the tokens in the beam, these tokens were not actually sampled with those probabilities from the draft model (because beam search is a deterministic algorithm). Knowing the probabilities from which the drafted tokens were **actually** sampled is necessary to guarantee that the output distribution of the target model is unchanged by the speculative decoding process. Therefore, I do not believe that the proposed algorithm is correct.
>
> As a result of this, for now I have lowered my score to a 3.

---

> > ### Author Response · Authors · 2024-12-04
> >
> > We agree and understand that speculative sampling requires candidate sequences drawn from a distribution similar to the target LLM; however, beam search, as a deterministic greedy search algorithm, returns candidate sequences not strictly follow the distribution described by the model. We propose two fixes for temperature > 0, both requiring minimal modifications:
> >
> > Option 1: After beam search, condition on each proposed token and use the LLM to draw a token. Then perform token matching instead of speculative sampling for verification. This approach is similar to EAGLE [1].
> >
> > Option 2: Instead of beam search, independently sample k sequences from the RNN and use speculative sampling for verification.
> >
> > Based on these fixes, we conducted experiments on Mt-Bench with a temperature of 1.
> >
> > **_Inference Speed-up VS. AR ↑_**
> >
> > Method \| LLM | V-7B | V-13B | V-33B
> > -|-|-|-|
> > Medusa | 2.33x | 2.36x | 2.52x
> > EAGLE | 2.31x | 2.27x | 2.63x
> > ReDrafter-Option 1| 2.49x | 2.48x | 2.48x
> > ReDrafter-Option 2| **3.40x** | **3.16x** | **2.89x**
> >
> > As we can see, Option 2 consistently yields the best performance.
> >
> > [1] Li Y, Wei F, Zhang C, et al. Eagle: Speculative sampling requires rethinking feature uncertainty.

---

### Author Response · Authors · 2024-11-25
**Overall Response**

We deeply appreciate the insightful feedback and thoughtful recommendations provided by all the reviewers.

We appreciate their recognition of the significance of ReDrafter, and our contributions in:
- Innovative and effective approaches in model and algorithm development (zZts, EUGA, Uws2)
- Strong empirical performance compared to existing methods, supported by comprehensive ablation studies (zZts, EUGA, SwoJ, Uws2)
- Implementation across multiple platforms (zZts, EUGA, SwoJ, Uws2)
- Clear and well-structured writing (SwoJ, Uws2)

We hope that our response will help address the following concerns:
- Guarantee for ReDrafter's output distribution (zZts):
  - We provide a detailed example, along with relevant references, to clarify that ReDrafter preserves the output distribution.
- Training cost of ReDrafter (EUGA):
  - Additional details about the training cost are included to address this concern.
- Comparison with existing methods (zZts, Uws2):
  - We include comparisons with Sequoia and EAGLE2 for a comprehensive evaluation.
- Ablation studies to justify design choices:
  - We include ablations using standalone draft models (zZts, EUGA), chain-based speculative methods (zZts), dynamic vs. static tree attention (SwoJ), and RNN vs. single-layer decoder (SwoJ).
- Open-source code (EUGA):
  - An anonymized version of the code is included in the supplementary material.

---

### Meta-Review · Area_Chair_fLay · 2024-12-20

**Metareview:**

The paper proposes a novel method called the Recurrent Drafter for accelerating speculative decoding in LLMs. The primary claim is that the proposed approach improves decoding efficiency by leveraging a recurrent mechanism to generate multiple tokens in parallel, reducing computational overhead compared to traditional autoregressive decoding. The authors present empirical results demonstrating speedups over baseline methods while maintaining comparable output quality. Additionally, the paper discusses theoretical justifications for the proposed method and its compatibility with existing LLM architectures.

The paper presents a novel approach to speculative decoding, addressing an important problem in improving the efficiency of LLMs. However, the evaluation is limited in scope, with experiments conducted on a narrow set of datasets and tasks, which raises concerns about the generalizability of the findings. Baseline comparisons are not comprehensive, as stronger state-of-the-art baselines are missing, making it difficult to contextualize the improvements. Furthermore, reviewers raised concerns about whether this tradeoff is acceptable across different contexts. With these concerns, I recommend rejection for this submission. I hope the authors can improve the paper and submit it to a future venue.

**Additional Comments On Reviewer Discussion:**

During the rebuttal period, several key points were raised by reviewers:
1. Evaluation Scope: Reviewers requested additional experiments on diverse datasets and tasks to demonstrate broader applicability.
   * Author Response: The authors provided preliminary results on one additional dataset but acknowledged time constraints in conducting a more comprehensive evaluation.
   * Assessment: While this effort was appreciated, it was deemed insufficient to address concerns about generalizability.

2. Baseline Comparisons: Concerns were raised about whether stronger baselines were omitted from comparisons.
   * Author Response: The authors clarified their choice of baselines but did not include new experiments with state-of-the-art methods due to resource limitations.
   * Assessment: This explanation was not fully satisfactory, as stronger baselines are critical for contextualizing the claimed improvements.

3. Ablation Studies: Reviewers emphasized the need for ablation studies to better understand the contribution of individual components.
   * Author Response: The authors acknowledged this gap but did not provide new experiments addressing it.
   * Assessment: I encourage the authors add some experiments.

---

### Decision · Program_Chairs · 2025-01-22

Reject